# High prevalence of non-communicable diseases among key populations enrolled at a large HIV prevention & treatment program in Kenya

**Dunstan Achwoka**[1]*, **Julius O. Oyugi**[1,2], **Regina Mutave**[1], **Patrick Munywoki**[3], **Thomas Achia**[1], **Maureen Akolo**[4], **Festus Muriuki**[4], **Mercy Muthui**[4], **Joshua Kimani**[1,4]

1 University of Nairobi Institute of Tropical and Infectious Diseases (UNITID), University of Nairobi, Nairobi, Kenya, 2 Department of Medical Microbiology and Infectious Diseases, University of Manitoba, Winnipeg, Canada, 3 School of Nursing and Public Health, Pwani University, Mombasa, Kenya, 4 University of Manitoba Research Group, Nairobi, Kenya

* dachwoka@students.uonbi.ac.ke

## Abstract

### Introduction

People Living with HIV (PLHIV) bear a disproportionate burden of non-communicable diseases (NCDs). Despite their significant toll across populations globally, the NCD burden among key populations (KP) in Kenya remains unknown. The burden of four NCD-categories (cardiovascular diseases, cancer, chronic respiratory diseases and diabetes) was evaluated among female sex workers (FSWs) and men who have sex with men (MSM) at the Sex Workers Outreach Program (SWOP) clinics in Nairobi Kenya.

### Methods

A retrospective medical chart review was conducted at the SWOP clinics among KP clients ≥15 years living with HIV enrolled between October 1, 2012 and September 30, 2015. The prevalence of the four NCD-categories were assessed at enrollment and during subsequent routine quarterly follow-up care visits as per the Ministry of Health guidelines. Prevalence at enrollment was determined and distributions of co-morbidities assessed using Chi-square and t-tests as appropriate during follow-up visits. Univariate and multivariate analysis were conducted to identify factors associated with NCD diagnoses.

### Results

Overall, 1,478 individuals' records were analyzed; 1,392 (94.2%) were from FSWs while 86 (5.8%) were from MSM over the three-year period. FSWs' median age was 35.3 years (interquartile range (IQR) 30.1–41.6) while MSM were younger at 26.8 years (IQR 23.2–32.1). At enrollment into the HIV care program, most KPs (86.6%) were at an early WHO clinical stage (stage I–II) and 1462 (98.9%) were on first-line anti-retroviral therapy (ART). A total of 271, 18.3% (95% CI: 16.4–20.4%), KPs living with HIV had an NCD diagnosis in

**Data Availability Statement:** Data are not publicly available because our study population focuses on an extremely vulnerable population, and our data

contain indirect identifiers that would potentially inadvertently identify our study subjects. In line with the Declaration of Helsinki, we are under ethical obligation to offer additional safeguards in protecting this population. Under protocol P258/09/2008, the Kenyatta National Hospital – University of Nairobi (KNH-UON) Ethics Research Committee has imposed restriction on the access of this data citing that sharing would be deemed to increase the risks or affect the safety or welfare of study participants. Data are available upon request the Secretary at KNH-UoN Ethics and Research Committee (contact via uonknh_erc@uonbi.ac.ke) for researchers who meet the criteria for access to confidential data.

**Funding:** The author(s) received no specific funding for this work.

**Competing interests:** The authors have declared that no competing interests exist.

their clinical chart records during the study period. Majority of these cases, 258 (95.2%) were noted among FSWs. Cardiovascular disease that included hypertension was present in 249/271, 91.8%, of KPs with a documented NCD. Using a proxy of two or more elevated blood pressure readings taken < 12 months apart, prevalence of hypertension rose from 1.0% (95% CI: 0.6–1.7) that was documented in the charts during the first year to 16.3% (95% CI: 14.4–18.3) in the third year. Chronic respiratory disease mainly asthma was present in 16/271, a prevalence of 1.1% (95% CI: 0.6–1.8) in the study population. Cancer in general was detected in 10/271, prevalence of 0.7% (95% CI: 0.3–1.2) over the same period. Interestingly, diabetes was not noted in the study group. Lastly, significant associations between NCD diagnosis with increasing age, body-mass index and CD4 + cell-counts were noted in univariate analysis. However, except for categories of $\geq$ BMI 30 kg/m$^2$ and age $\geq$ 45, the associations were not sustained in adjusted risk estimates.

## Conclusion

In Kenya, KP living with HIV and on ART have a high prevalence of NCD diagnoses. Multiple NCD risk factors were also noted against a backdrop of a changing HIV epidemic in the study population. This calls for scaling up focus on both HIV and NCD prevention and care in targeted populations at increased risk of HIV acquisition and transmission. Hence, KP programs could include integrated HIV-NCD screening and care in their guidelines.

## Introduction

The Global Burden of Disease Study 2017 ranked non-communicable diseases (NCDs) as the number one cause of mortality worldwide [1]. In Sub-Saharan Africa (SSA), NCDs now account for 37% of productivity losses overtaking communicable diseases and heralding an epidemiologic shift from infectious causes [2–4]. People Living with HIV (PLHIV) are disproportionately affected by the dual disease burden [5]. There is renewed focus to address NCDs among PLHIV [6, 7], yet key populations (KPs) who are an important segment of this population continue to lag behind in spite of their risky lifestyle choices.

The World Health Organization (WHO) identifies key populations as defined groups who, due to specific higher-risk behaviors, are at increased risk of HIV irrespective of the epidemic type or local context [8]. Often, key populations have legal and social issues related to their behavior that increase their vulnerability to HIV infection [8]. The Joint United Nations Programme on HIV and AIDS (UNAIDS) considers five key population groups as being particularly vulnerable to HIV infection namely: men who have sex with men (MSM), sex workers (SWs), people who inject drugs (PWIDs), transgender people and prisoners [9].

KPs, including those in SSA, carry a disproportionate burden of HIV; yet they have been under-represented wherever studied–particularly for HIV [10–12]. NCD burden of four main categories—cardiovascular diseases, diabetes mellitus, chronic respiratory diseases and cancer has been estimated in the general population living with HIV in SSA [5, 7, 13]. The four aforementioned categories are noted to contribute 80% of premature deaths [14]. Despite being excluded from many primary HIV surveillance systems, KPs account for 25% of new HIV infections in SSA [9, 15]. Further, risk factors such as harmful alcohol use, tobacco smoking and injecting drug use predispose KPs to both HIV infection and acquisition and progression of NCDs [11, 16].

Despite evidence on benefits of harm reduction among KPs, NCD-HIV care has received little attention [17]. In SSA countries, where KPs are recognized, NCD-HIV care packages have similarly lacked an emphasis on NCD-care [18]. Biomedical interventions aimed at NCD-HIV care such as cancer screening are considered as desirable rather than mandatory [19].

Using program data from a large key populations program in Nairobi, Kenya, this study sought to describe the NCD burden among two key population groups–female sex workers (FSWs) and MSM living with HIV enrolled in the SWOP clinics. For this paper, four main NCD categories—cardiovascular diseases, diabetes mellitus, chronic respiratory diseases and cancer recorded in the patient's clinical notes over a three-year period were evaluated.

## Materials and methods

### Study design and population

Data for this study were obtained from a medical chart review of clients enrolled in a large key populations' HIV prevention, care and treatment program in Nairobi Kenya. KPs enrolled in the Sex Workers Outreach Program (SWOP) included FSWs, and MSM. Those reached by SWOP team within Nairobi County are encouraged to enroll in the funded program that provides free, friendly, acceptable and accessible minimum package of HIV prevention and treatment services for sex workers as per the Ministry of Health guidelines [19]. Due to rampant stigma and discrimination in Kenya for MSM, this group started accessing available HIV prevention and treatment services within the last 10 years. Hence, they are under-represented in health care programs providing targeted, accessible, acceptable and free health care services. Medical charts of KPs living with HIV enrolled between October 2012 and September 2015 at all seven SWOP Drop-in Centers (DICEs), and on HIV treatment and care program spread across Nairobi County were included in the study. Specifically, medical charts of KPs aged 15 and above living with HIV (national antiretroviral therapy (ART) tools in Kenya classify ages $\geq$ 15 as adults), irrespective of ART initiation status were considered for analyses. Additional inclusion criteria included identification as either being a FSW or MSM. Criteria for exclusion in this study included being HIV-uninfected, being a PWID or transgender, missing age or age < 15 years and enrollment into the SWOP clinics before October 2012 or after September 2015.

### Study procedures and data collection

Medical records of those living with HIV enrolled in the program over the three-year period were abstracted during October and November 2018. At each of the seven constituent SWOP clinics, four trained abstractors collected data from clinic files of all HIV-infected individuals using a standardized data abstraction tool in MS-Excel. Details of each client's clinical encounter and follow-up visit during the study period were collected. Query scripts written in structured query language (SQL) were used to extract ART care data-variables contained in the national Ministry of Health (MOH) forms. All SWOP clinics utilize the nationally approved ART electronic medical records systems that contains the national MOH ART patient care forms. Variables that fell outside the purview of the query scripts were manually extracted and double-entered into the MS-Excel abstraction tool for validation.

The team worked under the supervision of a data manager and program manager who verified abstracted data for completeness and accuracy on a daily basis to assure data quality. Data was transmitted encrypted on a daily basis and stored at a server at the central SWOP office in Nairobi. All computers used for abstraction and storage were password protected and access

limited to only the data management team. Data were cleaned and subsequently imported to STATA 15 (STATA Corporation, Texas USA) for data analysis.

## Statistical analyses

This study's analyses included medical chart records of two key population typologies: FSWs and MSM who were HIV-infected. Descriptive statistics were used to compute means, standard deviations (SD) and 95% confidence intervals (CI) for numerical variables as well as frequencies for ordinal and categorical variables. The baseline characteristics of the study participants were compared by KP type using appropriate statistics (chi-square or fisher's exact test as necessary for categorical variables and t-tests for continuous variables). Depending on the backbone antiretroviral drug molecule—nucleoside reverse transcriptase inhibitor (NRTI) or protease inhibitor (PI), antiretroviral treatment regimens were classified as either being first line or second line. The main outcome was any NCD derived from report of cardiovascular disease, diabetes mellitus, chronic respiratory diseases or cancer at enrollment and during HIV treatment and care (study period). Prevalence of the specific NCDs and any NCDs was calculated stratified by KP typology for a range of population characteristics. Univariate and multivariable logistic regression were conducted to identify factors associated with NCD diagnoses. An automated stepwise backward logistic regression approach was used to identify independent predictors of NCDs retaining variables with a p-value of 0.2 from the univariate analysis. Age, gender, alcohol use and smoking were considered apriori as potential confounders and included in the final multivariable model. Collinearity and interaction of the variables was assessed. A sensitivity check through an analysis that included missing data confirmed the assumption that data was missing at random. Missing data were not imputed. Analyses on non-clinical measures presented were based on self-reported data.

## Ethical considerations

The analyses of these routine HIV treatment and care data from Nairobi County SWOP clinics was approved by the Kenyatta National Hospital, University of Nairobi Ethics Review Committee; (KNH UON ERC P258/09/2008) and as part of a nested study (KNH UON ERC P720/10/2018). Prior to accessing the required data for this study, the data manager de-identified the patients' clinical charts creating anonymity as way of maintaining confidentiality. Upon enrollment into SWOP, all patients provided informed consent to clinical data collection that allowed use of their clinic charts to inform HIV prevention, care and treatment in Kenya. Annual approvals were granted by the Kenyatta National Hospital, University of Nairobi Ethics research committee upon satisfactory review of annual study progress reports under protocol P258/09/2008. Being of a secondary nature, there was no human subject interface during the conduct of this study.

## Results

### Baseline characteristics of the study population

Clinical encounters from October 2012 to September 2015 were analyzed for 1,478 clients. Among these individuals, 1,392 (94.2%) were FSWs, while 86 (5.8%) were MSM. Overall, majority of medical records were obtained from two SWOP facilities: Majengo (24.8%) and SWOP City (22.4%). Majengo facility served over a quarter (26.3%) of FSWs while slightly over two thirds (67.4%) of MSM sought services at SWOP City. The rest of the five SWOP clinics constituted the slightly over a half of the medical records (52.8%). Median age of FSWs was 35.3 years (interquartile range (IQR) of 30.1–41.6) and that of MSM was 26.8 years

Table 1. Baseline characteristics of key populations living with HIV attending SWOP clinics by typology, 2012–2015 (N = 1,478).

| Characteristics | Total (N = 1,478) | | Key Population Typology | | | |
| --- | --- | --- | --- | --- | --- | --- |
| | | | FSW (n = 1,392) | | MSM (n = 86) | |
| | No. | % | n | % | n | % |
| Age (years) | | | | | | |
| Mean [SD] | 35.8 | [8.5] | 36.2 | [8.4] | 28.2 | [6.7] |
| 15–25 | 138 | 9.3 | 102 | 7.3 | 36 | 41.9 |
| 25–34 | 601 | 40.7 | 565 | 40.6 | 36 | 41.9 |
| 35–44 | 512 | 34.6 | 501 | 36 | 11 | 12.8 |
| 45+ | 224 | 15.1 | 221 | 15.9 | 3 | 3.5 |
| Facility | | | | | | |
| Donholm | 125 | 8.5 | 118 | 8.5 | 7 | 8.1 |
| Majengo | 367 | 24.8 | 367 | 26.3 | N/A[1] | |
| SWOP City | 331 | 22.4 | 273 | 19.6 | 58 | 67.4 |
| Kariobangi | 193 | 13.1 | 181 | 13 | 12 | 14 |
| Kawangware | 201 | 13.6 | 196 | 14.1 | 5 | 5.8 |
| Langata | 111 | 7.5 | 107 | 7.7 | 4 | 4.7 |
| Thika Road | 150 | 10.2 | 150 | 10.8 | N/A | |
| Marital Status | | | | | | |
| Married | 220 | 14.9 | 202 | 14.5 | 18 | 20.9 |
| Widowed | 72 | 4.9 | 71 | 5.1 | 1 | 1.2 |
| Divorced | 484 | 32.8 | 479 | 34.4 | 5 | 5.8 |
| Single | 689 | 46.6 | 627 | 45.0 | 62 | 72.1 |
| WHO Stage at Enrolment | | | | | | |
| I-II | 1279 | 86.6 | 1198 | 86.1 | 81 | 94.2 |
| III—IV | 123 | 8.3 | 121 | 8.7 | 2 | 2.3 |
| Undocumented | 76 | 5.1 | 73 | 5.2 | 3 | 3.5 |
| CD4 T-Cell Count | | | | | | |
| <200 | 257 | 17.4 | 251 | 18 | 6 | 7 |
| 200–349 | 405 | 27.4 | 369 | 26.5 | 36 | 41.9 |
| 350–499 | 326 | 22.1 | 305 | 21.9 | 21 | 24.2 |
| 500+ | 404 | 27.3 | 388 | 27.9 | 16 | 18.6 |
| Undocumented | 86 | 5.8 | 79 | 5.7 | 7 | 8.1 |
| Antiretroviral Treatment Regimen | | | | | | |
| First line (NRTI based) | 1462 | 98.9 | 1376 | 98.9 | 86 | 100.0 |
| Second line (PI based) | 16 | 1.1 | 16 | 1.2 | 0 | 0 |

[1]N/A -Not applicable since the facility was an FSWs only clinic and did not enroll MSM during the study period.

(IQR 23.2–32.1). Close to half (46.6%) of all KPs were single, 32.8% were divorced, 14.9% married and 4.9% were widowed. The proportion of FSWs that was single was lower than that of MSM (45.0% vs 72.1% respectively) (Table 1).

At entry into SWOP, 97.7% of KPs in this study cohort were HIV infected. Sixteen clients across both KP typologies (11 FSWs and 5 MSM), initially HIV-uninfected at entry into SWOP, seroconverted during follow up. Seroconversion among FSWs was 0.8% while that of MSM was 5.8% (results not shown). At the time of enrollment into HIV care, most KPs (86.6%) were at an early (stage I–II) WHO clinical stage. Less than a fifth (17.4%) of all clients had a CD4 T-cell count of less than 200 cells/mm$^3$. Over a quarter (27.9%) of FSWs and 18.6%

of MSM had a CD4 count of 500 cells/mm$^3$ and above. Nearly all (98.9%) clients enrolled were initiated on a first line antiretroviral regimen (Table 1).

## Prevalence of NCDs among key populations living with HIV

A total of 271, 18.3% (95% CI: 16.4–20.4), KPs living with HIV had an NCD diagnosis in their clinical chart records. The vast majority, 95.2% (258 cases) of all the NCDs were from the FSWs. About a third (33.9%) of the NCDs were reported from Majengo, where 25.1% (95% CI: 20.7–29.8) of FSWs at this facility had an NCD diagnosis. A similar proportion of MSM had an NCD diagnosis in their medical charts at Kariobangi 25.0%, (95% CI: 5.5–57.2). KPs aged between 35–44 years had the highest number of NCD diagnoses 108/271, 21.1% (95% CI: 17.6–24.9) (Table 2). FSWs' NCD prevalence rose steadily with age, 7.8% (95% CI: 3.5–14.9) among the under 25 years of age to 33.0% (95% CI: 26.9–39.7) among those aged 45 years and above. MSM NCD prevalence was highest among those aged 35–44 years, 18.2% (95% CI: 2.3–51.8) and lowest among those aged between 25 and 34 years 11.1% (95% CI: 3.1–26.1) (Fig 1a).

At enrollment into HIV care, 34/271, 12.5%, KPs living with HIV and with an NCD diagnosis had advanced disease with a CD4 of less than 200 cells/mm$^3$. All 34, were FSWs and had an NCD prevalence of 13.5% (95% CI: 9.6–18.4). Thirty one percent of KP clients living with HIV and diagnosed with an NCD had an enrollment CD4 of $\geq$ 500 cells/mm$^3$. For both KP typologies, NCD prevalence for the $\geq$ 500 cells/mm$^3$ CD4 category was close to a fifth; 20.8% (95% CI; 16.9–25.3) for FSWs and 18.8% (95% CI; 4.1–45.6) for MSM respectively. Nearly all, 268/271, 98.9%, of KPs with an NCD diagnosis were currently on an NRTI-based first line ART regimen. Three FSWs were on a protease inhibitor (PI) based second line regimen and had an NCD prevalence of 18.8% (95% CI: 4.1–45.7). Two thirds, (66.8%) of KPs living with HIV and with an NCD diagnosis had a body mass index (BMI) range of either being overweight 83/271, 30.6% or obese 98/271, 36.2%. Prevalence of NCD among overweight FSWs was 19.6% (95% CI: 15.879–23.8). Overweight MSM had an NCD prevalence of 25.0% (95% CI: 5.5–57.2) (Table 2).

Most KP clients living with HIV and an NCD diagnosis, 116/271, 42.8% reported a mixed profile of sexual partners that included both regular and casual sexual clients as well as an intimate sexual partner. Among FSWs with a mixed profile of partners, NCD prevalence was 18.8% (95% CI: 15.7–22.1). Close to two fifths (38.5%) of HIV-infected MSM with an NCD diagnosis had a casual client and an NCD prevalence of 16.7% (95% CI: 5.6–34.7). Vast majority (99.4%) of both FSWs and MSM reported consistent use of condoms with casual clients (Table 2).

Almost two thirds, 180/271, 66.4%, of KPs living with HIV and with an NCD diagnosis consumed alcohol with 18/180, 10%, screening positive on the CAGE tool for excessive drinking. NCD prevalence among FSWs and MSM who screened positive for excessive drinking was 15.3% (95% CI: 8.8–24.0) and 37.5% (95% CI: 8.5–75.5) respectively. However, a quarter, 70/271, 25.8%, of KPs living with HIV and with an NCD diagnosis did not take alcohol. A majority of KPs living with HIV and with an NCD diagnosis, 160/271, 59.0%, did not smoke. Close to a tenth, 23/271, 8.4%, smoked tobacco cigarettes; all were FSWs. NCD prevalence for FSWs who smoked more than one pack a day was 8.0% (95% CI: 1.0–26.0). Drug use was reported among 5.9% of KPs living with HIV and with an NCD diagnosis cohort, all being FSWs. NCD prevalence among 16 FSWs who reported drug use was 13.2% (95% CI: 7.8–20.6) (Table 2).

**Cardiovascular disease.** Among KPs living with HIV and with a documented NCD, 249/271, 91.8%, had a form of cardiovascular disease (CVD) that included hypertension. CVD was more frequent in FSWs than MSM 17.0% (95% CI: 15.1–19.1) vs 14.0% (95% CI: 7.4–23.1) respectively. Among FSWs, CVD prevalence was lowest in the under 25 years age band 5.9%

**Table 2. Prevalence of Non-Communicable Diseases (NCDs) among key populations living with HIV by selected characteristics at SWOP clinics, 2012–2015.**

| Characteristics | Categories | N | Overall | | FSW | | MSM | |
|---|---|---|---|---|---|---|---|---|
| | | | n | % [95% C.I] | n | % [95% C.I] | n | % [95% C.I] |
| Facility | Donholm | 125 | 19 | 15.2 [9.4–22.7] | 18 | 15.3 [9.3–23.0] | 1 | 14.3 [0.4–57.9] |
| | Majengo | 367 | 92 | 25.1 [20.7–29.8] | 92 | 25.1 [20.7–29.8] | | N/A |
| | SWOP city | 331 | 58 | 17.5 [13.6–22.1] | 49 | 18.0 [13.6–23.0] | 9 | 15.5 [7.4–27.4] |
| | Kariobangi | 193 | 27 | 14.0 [9.4–19.7] | 24 | 13.3 [8.7–19.1] | 3 | 25.0 [5.5–57.2] |
| | Kawangware | 201 | 32 | 15.9 [11.2–21.7] | 32 | 16.3 [11.4–22.3] | 0 | 0 |
| | Langata | 111 | 4 | 3.6 [1.0–9.0] | 4 | 3.74 [1.0–9.3] | 0 | 0 |
| | Thika Road | 150 | 39 | 26.0 [19.2–33.8] | 39 | 26.0 [19.2–33.8] | | N/A |
| Age bands (years) | <25 | 138 | 15 | 10.9 [6.2–17.3] | 8 | 7.8 [3.5–14.9] | 7 | 19.4 [8.2–36.0] |
| | 25–34 | 601 | 75 | 12.5 [9.9–15.4] | 71 | 12.6 [9.9–15.6] | 4 | 11.1 [3.1–26.1] |
| | 35–44 | 512 | 108 | 21.1 [17.6–24.9] | 106 | 21.2 [17.7–25.0] | 2 | 18.2 [2.3–51.8] |
| | 45+ | 224 | 73 | 33.0 [22.5–35.8] | 73 | 33.0 [26.9–39.7] | 0 | 0 |
| Body Mass Index (kg/m$^2$) | <18.5 | 80 | 8 | 10.0 [4.4–18.8] | 8 | 11.3 [5.0–21.0] | 0 | 0 |
| | 18.5–24.9 | 647 | 79 | 12.2 [9.8–15.0] | 69 | 11.8 [9.3–14.7] | 10 | 15.9 [7.9–27.3] |
| | 25–29.9 | 420 | 83 | 19.8 [16.1–23.9] | 80 | 19.6 [15.9–23.8] | 3 | 25.0 [5.5–57.2] |
| | 30+ | 301 | 98 | 32.6 [27.3–38.2] | 98 | 32.8 [27.5–38.4] | 0 | 0 |
| Sex partner type | Casual client | 282 | 51 | 18.1 [13.8–23.1] | 46 | 18.3 [13.7–23.6] | 5 | 16.7 [5.6–34.7] |
| | Regular client | 78 | 8 | 10.3 [4.5–19.2] | 8 | 11.4 [5.1–21.3] | 0 | 0 |
| | Regular; Casual clients + Partner | 620 | 116 | 18.7 [15.7–22.0] | 114 | 18.8 [15.7–22.1] | 2 | 15.4 [1.9–45.5] |
| | Regular partner | 15 | 2 | 13.3 [1.7–40.5] | 1 | 50.0 [1.3–98.7] | 1 | 7.7 [0.2–36.0] |
| | Undocumented | 482 | 94 | 19.5 [16.06–23.3] | 89 | 19.35 [15.8–23.3] | 5 | 22.7 [7.8–45.4] |
| Condom use | No | 10 | 1 | 10 [0.3–44.5] | 1 | 25.0 [0.6–80.6] | 0 | 0 |
| | Yes | 1102 | 194 | 17.6 [15.4–20.0] | 186 | 17.9 [15.6–20.3] | 8 | 12.9 [5.7–23.9] |
| | Undocumented | 365 | 76 | 20.8 [16.8–25.4] | 71 | 20.5 [16.3–25.1] | 5 | 27.9 [9.7–53.5] |
| Alcohol consumption (Cage per day) | 0 | 419 | 70 | 16.7 [13.2–20.6] | 67 | 17.1 [13.5–21.2] | 3 | 10.7 [2.3–28.2] |
| | 1 | 871 | 162 | 18.6 [16.1–21.3] | 156 | 18.7 [16.1–21.5] | 6 | 15.4 [5.9–30.5] |
| | 4 | 106 | 18 | 17.0 [10.4–25.5] | 15 | 15.3 [8.8–24.0] | 3 | 37.5 [8.5–75.5] |
| | Undocumented | 79 | 21 | 26.6 [17.3–37.7] | 20 | 29.4 [19.0–41.7] | 1 | 9.1 [2.3–41.3] |
| Smoking (Packs per day) | 0 | 920 | 160 | 17.4 [15.0–20.0] | 147 | 17.5 [15.0–20.2] | 13 | 17.1 [9.4–27.5] |
| | 1 | 87 | 21 | 24.1 [15.6–34.5] | 21 | 25.0 [16.2–35.6] | 0 | 0 |
| | 2 | 29 | 2 | 6.9 [0.9–22.8] | 2 | 8.0 [1.0–26.0] | 0 | 0 |
| | Undocumented | 444 | 88 | 19.8 [16.2–23.8] | 88 | 20.0 [16.3–24.0] | 0 | 0 |
| Drugs use | No | 1337 | 250 | 18.7 [16.7–20.9] | 237 | 18.9 [16.8–21.2] | 13 | 16.1 [8.8–25.9] |
| | Yes | 125 | 16 | 12.8 [7.5–20.0] | 16 | 13.2 [7.8–20.6] | 0 | 0 |
| CD 4 cells/mm$^3$ | <200 | 258 | 34 | 13.2 [9.3–18.0] | 34 | 13.5 [9.6–18.4] | 0 | 0 |
| | 200–349 | 404 | 82 | 20.3 [16.4–24.5] | 76 | 20.6 [16.6–25.1] | 6 | 16.7 [6.4–32.8] |
| | 350–499 | 325 | 67 | 20.6 [16.3–25.4] | 63 | 20.6 [16.3–25.6] | 4 | 19.1 [5.5–41.9] |
| | 500+ | 404 | 84 | 20.8 [16.9–25.1] | 81 | 20.8 [16.9–25.3] | 3 | 18.8 [4.1–45.6] |
| ART Regimen | First line | 1464 | 268 | 18.3 [16.4–20.4] | 255 | 18.5 [16.5–20.7] | 13 | 15.1 [8.3–24.5] |
| | Second line | 16 | 3 | 18.8 [4.1–45.7] | 3 | 18.8 [4.1–45.7] | | 0 |

FSWs, female sex workers; MSM, Men who have sex with men; ART, antiretroviral therapy; N, total number of participants in each category; n, number of participants with NCD; %, percentage with NCDs

(95% CI: 2.2–12.4) and rose across age bands to 31.7% (95% CI: 25.6–38.3) in those aged 45 years and above. Among MSM, the highest CVD prevalence was in the under 25 years age band while the lowest was in the 25–34 years age band 19.4% (95% CI: 8.2–36.0) vs 11.1% (95% CI: 3.1–26.1) (Fig 1b).

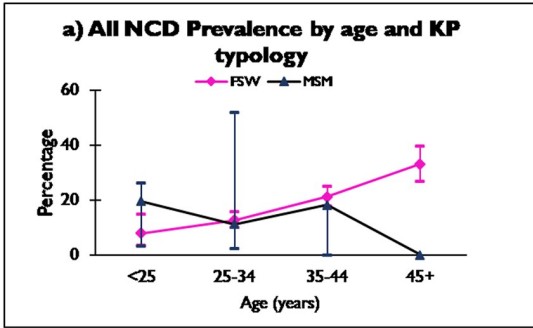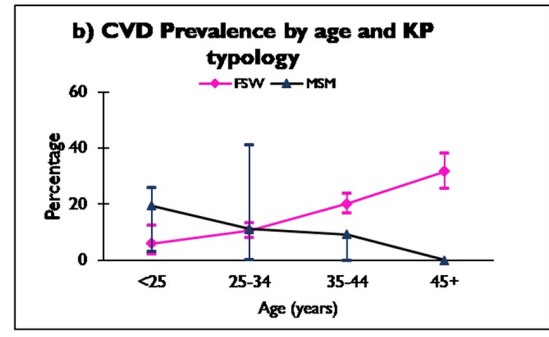

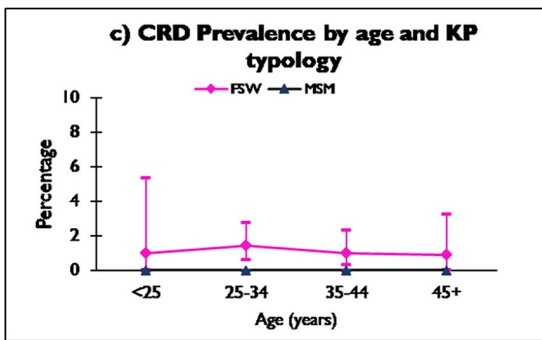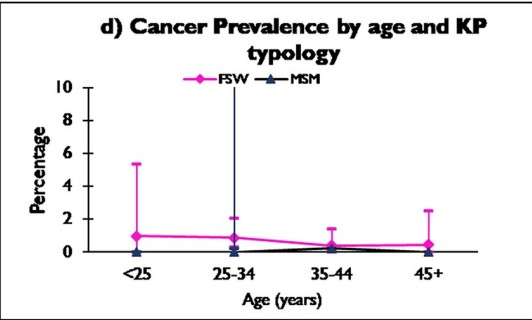

**Fig 1. a.** All NCD prevalence by age and KP typology. **b.** CVD prevalence by age and KP typology. **c.** CRD prevalence by age and KP typology. **d.** Cancer prevalence by age and KP typology.

Prevalence of hypertension as documented in reviewed KP medical records was 1.0% (95% CI: 0.6–1.7) with all cases being from FSWs. When two or more elevated blood pressure readings taken <12 months apart were considered, prevalence of elevated blood pressure was 16.3% (95% CI: 14.4–18.3). Proxy measure of hypertension was based on the Seventh Joint National Commission on hypertension (JNC 7) definition. Elevated blood pressure readings were more common among FSWs than MSM 16.5% (95% CI: 14.5–18.6) vs 14.0 (95% CI: 7.4–23.1) respectively. While serial elevated blood pressure readings were detected in 233/249 KP medical records, only 15/249 had a documented diagnosis of hypertension. Other CVD diagnoses such as atherosclerotic heart disease and congestive heart failure were made in 5/249 cases of CVD with a prevalence of 0.3% (95% CI: 0.1–0.8) (Table 3).

**Table 3. Prevalence of Non-Communicable Diseases (NCDs) among key populations living with HIV at SWOP clinics in Nairobi, Kenya, 2012–15.**

| NCD type | Total (N = 1478) | | FSW (n = 1392) | | MSM (n = 86) | |
|---|---|---|---|---|---|---|
| | *n* | *% [95% C.I]* | *n* | *% [95% C.I]* | *n* | *% [95% C.I]* |
| Any | *271* | 18.3[16.4–20.4] | *258* | 18.5 [16.5–20.7] | *13* | 15.1[8.3–24.5] |
| Cardiovascular Disease (CVD) | 249 | 16.9 [15.0–18.9] | 237 | 17.0 [15.1–19.1] | 12 | 14.0 [7.4–23.1] |
| Elevated Blood Pressure[1] | 233 | 16.3 [14.4–18.3] | 221 | 16.5 [14.5–18.6] | 12 | 14.0 [7.4–23.1] |
| Hypertension Diagnosis[2] | 15 | 1.0 [0.6–1.7] | 15 | 1.1 [0.6–1.8] | 0 | 0 |
| Other CVD Diagnoses | 5 | 0.3 [0.1–0.8] | 5 | 0.4 [0.1–0.8] | 0 | 0 |
| Chronic Respiratory Disease | 16 | 1.1[0.6–1.8] | 16 | 1.2 [0.7–1.9] | 0 | 0 |
| Cancer | 10 | 0.7 [0.3–1.2] | 9 | 0.7 [0.3–1.2] | 1 | 1.2 [0.0–6.3] |

[1]Elevated blood pressure is calculated based on two elevated blood pressure readings taken <12 months apart in line with JNC 7 definition;

[2]Hypertension diagnosis denotes documented hypertension diagnosis found in medical charts;

[3] Diabetes mellitus is not shown on the table since no record of the condition was found in the entire study population

**Chronic respiratory disease.** A total of 16/271 medical records of KPs living with HIV reviewed were found to have a documented chronic respiratory disease (CRD). Overall prevalence of CRD was 1.1% (95% CI: 0.6–1.8). All cases reviewed were documented cases of asthma among FSWs (Table 3). The highest CRD prevalence was observed among FSWs aged 25–34 years 1.4% (95% CI: 0.6–2.8) (Fig 1c).

**Cancer.** A total of 10/271 records of KPs living with HIV were found to have documentation of a cancer diagnosis. Overall prevalence of cancer was estimated at 0.7% (95% CI: 0.3–1.2). Nine of the ten cancer diagnoses were of cervical cancer among FSWs. Cervical cancer diagnoses were made at two SWOP facilities–Donholm and Kawangware. The type of cancer was not specified for the one cancer diagnosis made on an MSM. Although 8 of the 10 cancer cases reported a mixed profile (regular clients, casual clients and a regular partner) for their sexual partner, all were found to have consistent condom use (results not shown). Majority of cervical cancer diagnoses (5/9) were made among the 25–34 years age-band. The one MSM who had a cancer diagnosis was in the 35–44 years age band (Fig 1d).

**Diabetes mellitus.** In this cohort of KP clients living with HIV, none of the FSWs or MSM were found to have a documented diagnosis of diabetes mellitus in their medical records.

## Predictors of NCD among key populations living with HIV at SWOP

On univariate analysis, increased age among KPs living with HIV was associated with an NCD diagnosis. The unadjusted odds ratio (OR) for 35–44 years age band was 2.19 (95% CI: 1.23–3.90) (p = 0.008) and that of 45 years and above 3.96 (2.17–7.26) (p = 0.001). Increased body mass index (BMI) was associated with an NCD diagnosis among KPs living with HIV. A BMI of 25–29.9 kg/m$^2$ (overweight) among the HIV- infected KP was associated with an OR 2.22 (95% CI: 1.03–4.78) (p = 0.042) while those with a BMI of $\geq$ 30 (obese) had an OR 4.34 (95% CI: 2.01–9.38) (p = 0.001). Similarly, increasing CD4 cells/mm$^3$ was associated with a documented NCD diagnosis. Odds among CD4 counts of 200–349 cells/mm$^3$ was OR 1.67 (95% CI: 1.08–2.57) (p = 0.022). CD4 counts of $\geq$500 cells/mm$^3$ had an OR 1.72 (1.11–2.66) (p = 0.014) (Table 4).

Other predictive variables considered in the univariate analyses (sex, smoking, alcohol use, drug use, current ART regimen, sexual partner profile, and previous history of TB) were all not significant at a p-value of 0.2. Even though increased age, BMI and CD4 were associated with NCD diagnosis in unadjusted analyses, significant association with NCD diagnosis in the adjusted analyses remained only for categories of $\geq$ BMI 30 kg/m$^2$, and ages $\geq$ 45 years (borderline statistically significant) (Table 4).

## Discussion

This study described the burden of NCDs among key populations (KPs) living with HIV enrolled at a large prevention and treatment program in Nairobi, Kenya. It determined prevalence of four NCD conditions: cardiovascular diseases, diabetes mellitus, chronic respiratory illnesses and any form of cancer among two KP typologies–FSWs and MSM living with HIV at seven SWOP clinics in Nairobi. Further, distribution of prominent NCD risk factors and associated correlates among the two KP typologies were explored. This study comes against a backdrop of a rising NCD epidemic in SSA among PLHIV in the context of an evolving HIV epidemic with high unmet response for KPs [20, 21]. A high overall prevalence of any of the four NCDs (18.3%) was found among both HIV-infected FSWs and MSM. Despite a heightened impetus to refocus on populations at increased risk for both NCDs and HIV infection, studies among key populations living with HIV remain rare [22]. That notwithstanding,

**Table 4. Risk factors for NCDs among key populations living with HIV at SWOP Clinics in Nairobi, Kenya, 2012–15.**

| Characteristics | Categories | N | Any NCD | | Unadjusted Odds Ratio | | Adjusted Odds Ratio | |
|---|---|---|---|---|---|---|---|---|
| | | | *n* | *% [95% C.I]* | *OR [95% CI]* | *p-value* | *OR [95% CI]* | *p-value* |
| Age in years | 15–25 | 138 | 15 | 10.9 [6.2–17.3] | Reference | | Reference | |
| | 25–34 | 601 | 75 | 12.5 [9.9–15.4] | 1.17 [0.65–2.11] | 0.602 | 0.87 [0.45–1.67] | 0.672 |
| | 35–44 | 512 | 108 | 21.1 [17.6–24.9] | 2.19 [1.23–3.90] | 0.008 | 1.53 [0.79–2.95] | 0.209 |
| | 45+ | 224 | 73 | 32.6 [26.5–39.2] | 3.96 [2.17–7.26] | 0.001 | 2.10 [0.98–4.49] | 0.055 |
| Sex | Female | 1392 | 258 | 18.5 [16.5–20.7] | Reference | | Reference | |
| | Male | 86 | 13 | 15.1[8.3–24.5] | 0.78 [0.43–1.43] | 0.428 | 1.39 [0.69–2.79] | 0.354 |
| Smoking | No | 918 | 160 | 17.4 [15.0–20.0] | Reference | | Reference | |
| | Yes | 116 | 23 | 19.8 [13.0–28.3] | 1.17 [0.72–1.91] | 0.524 | 1.17 [0.67–2.04] | 0.583 |
| Alcohol Use | No | 420 | 70 | 16.7 [13.2–20.6] | Reference | | Reference | |
| | Yes | 979 | 180 | 18.4 [16.0–21.0] | 1.13 [0.83–1.53] | 0.442 | 0.95 [0.66–1.38] | 0.794 |
| Drug Use | No | 1336 | 250 | 18.7 [16.7–20.9] | Reference | | Reference | |
| | Yes | 125 | 16 | 12.8 [7.5–20.0] | 0.64 [0.37–1.10] | 0.104 | 1.25 [0.66–2.39] | 0.5 |
| Body Mass Index (kg/m$^2$) | <18.5 | 80 | 8 | 10.0 [4.4–18.8] | Reference | | Reference | |
| | 18.5–24.9 | 647 | 79 | 12.2 [9.8–15.0] | 1.25 [0.58–2.70] | 0.566 | 1.21 [0.49–3.00] | 0.680 |
| | 25–29.9 | 420 | 83 | 19.8 [16.1–23.9] | 2.22 [1.03–4.78] | 0.042 | 1.73 [0.69–4.39] | 0.246 |
| | 30+ | 301 | 98 | 32.6 [27.3–38.2] | 4.34 [2.01–9.38] | 0.001 | 2.87 [1.11–7.41] | 0.029 |
| ART Regimen | NRTI based | 1462 | 268 | 18.3 [16.4–20.4] | Reference | | N/A | |
| | PI based | 16 | 3 | 18.8 [4.1–45.6] | 1.03 [0.29–3.63] | 0.966 | | |
| CD4 | <200 | 257 | 34 | 13.2 [9.3–18.0] | Reference | | Reference | |
| | 200–349 | 405 | 82 | 20.3 [16.4–24.5] | 1.67 [1.08–2.57] | 0.022 | 1.42 [0.86–2.35] | 0.171 |
| | 350–499 | 326 | 67 | 20.6 [16.3–25.4] | 1.70 [1.08–2.66] | 0.021 | 1.25 [0.72–2.16] | 0.431 |
| | 500+ | 404 | 84 | 20.8 [16.9–25.1] | 1.72 [1.11–2.66] | 0.014 | 1.08 [0.63–1.86] | 0.780 |
| Previous TB history | No | 1471 | 269 | 18.3 [16.3–20.4] | Reference | | N/A | |
| | Yes | 6 | 1 | 16.7 [0.4–64.1] | 0.89 [0.10–7.68] | 0.918 | | |
| Sex Partner Type | Casual client | 282 | 51 | 18.1 [13.8–23.1] | Reference | | N/A | |
| | Regular client | 78 | 8 | 10.3 [4.5–19.2] | 0.51 [0.23–1.14] | 0.103 | | |
| | Regular client + Partner +Casual Client | 621 | 116 | 18.7 [15.7–22.0] | 1.04 [0.72–1.50] | 0.831 | | |
| | Regular Partner | 15 | 2 | 13.3 [1.7–40.5] | 0.70 [0.15–3.18] | 0.641 | | |

NCD: Non-communicable diseases. CI: Confidence interval.

systematic reviews outside SSA suggest that sexual minorities exhibit higher rates of NCDs [11]. Contrastingly, study findings from SSA point to comparably lower prevalence rates of NCDs (4.7%, 11.5% and 21.2%) among general population PLHIV clients [5, 13, 23, 24].

Study findings from concentrated HIV epidemics that are driven by an increased prevalence of HIV among key populations, point to high prevalence of NCDs among PLHIV [25]. In a Cambodian study among PLHIV, close to half (47.8%) of total study participants had one or more NCDs with 75% unaware of their disease condition prior to the study [26]. A recent modeling report from Kenya estimated 33% of HIV negative individuals and 36% of PLHIV to have at least one NCD. Further, prevalence of hypertension among HIV negative individuals was projected to grow from 19.9% in 2018 to 23% in 2035. This was in stark comparison to a growth from 29.9% to 37.4% among PLHIV over a similar period [27]. These findings enunciate the excess NCDs burden among key populations and point to the need for routine active screening to increase early identification.

Evidence around cardiometabolic risk factors for NCDs among PLHIV is mixed. While some studies suggest that HIV infection is associated with lower BMI, triglycerides and blood

pressure readings [28, 29], several others point to an increased prevalence of hypertension and obesity [5, 7, 30]. There are mixed associations with ART use on the prevalence of NCDs with some studies suggesting no associations with hypertension [23, 31] while others finding an increased odds for hypertension, dyslipidemia and other cardiovascular conditions [29, 30, 32]. Although chronic immune activation contributes to increased hypertension among PLHIV, the inflammatory milieu is poorly understood [33]. ART associated endothelial dysfunction [34], increasing age and longevity on ART treatment have also been associated with increased prevalence of NCDs among PLHIV [23, 30, 32]. In this study, close to two thirds of KPs living with HIV who had an NCD diagnosis, were either obese or overweight. A majority were FSWs. A higher prevalence of NCD diagnoses was observed with increased age.

This study found 1.0% prevalence of hypertension from clinical records. Using this study's proxy for hypertension of two or more blood pressure readings taken less than 12 months apart, the prevalence of hypertension rose to 16.3%. Further, this study found a low prevalence of other CVD diagnoses (0.3%). Similar discrepancies have been reported in other studies in SSA [13, 35]. In a South African study, prevalence of hypertension was higher during the day of the interview than when compared to both self-report and client records [35]. While the underdiagnosis in this latter study may be attributed to 'white coat hypertension', this study was considered as having a much more robust estimate of hypertension prevalence. However, while other studies reported high prevalence of other CVD diagnoses, isolating confounders of central nervous system (CNS) infections especially among ART naïve immunosuppressed clients proved difficult [36]. In this study, low prevalence of other CVD diagnoses, chronic respiratory diseases (1.1%), and diabetes mellitus (no cases) could have been attributed to absence of routine screening against a backdrop of a non-integrated NCD and HIV care system [37, 38].

ART treatment for KPs has generally followed a similar trajectory to that of the general population. [8]. Expanded ART eligibility over recent years has seen KPs with higher CD4 levels initiating ART through the Test and Treat platform. In this study, less than a fifth of KPs had advanced disease (less than CD4 count of 200 cells/mm$^3$) demonstrating benefits of the adopted test and treat strategy. A significant association between an NCD diagnosis and CD4 measurement was not found. While this is similar to findings in SSA [35], studies in high income countries have found associations between NCDs and a detectable viral load [39]. The underdiagnoses of NCDs that was common in this study, may have contributed to the absence of an association between NCD diagnosis and CD4 measurements.

Key populations engage in risky behavior that increase their risk for NCDs. Studies have documented harmful consumption of alcohol, smoking tobacco, illicit drug use, and risky sexual behaviors among KPs as factors that increase their risk for NCDs [40–42]. A systematic review among KPs in SSA, found a median prevalence of alcohol misuse based on AUDIT/CAGE of 32.8%; and that of illicit drug use ranging from 0.1% to 97.1% for injecting drug users [16]. Difficult social conditions, including criminalization of sex work, uneven coverage of biomedical interventions and stigma impact negatively on NCDs among KPs [11, 20, 21]. In this study, a high prevalence of NCDs (18.7%) was observed among FSWs who screened positive for excessive drinking. Among MSM, the prevalence was close to 2.5 times as high, albeit drawn from a small sample size. A tenth of the KPs smoked and had an NCD prevalence of less than 10%. Illicit drug use was reported by about one in twenty KPs, who also reported a low NCD prevalence. Close to a half of all KPs living with HIV who had an NCD diagnosis reported a mixed profile of sexual partners but with near universal condom use. Although sex work remains criminalized in this study setting, KPs receiving care at SWOP had good access to biomedical interventions including prevention and treatment services at both fixed clinics and through peer outreach models.

A recent systematic review on the Global burden of disease indicates that cancer cases have increased in developing countries of SSA and contribute significantly to years of life lost to disability [43]. Utilizing registry data from Malawi, an earlier study pointed to a high burden of AIDS defining cancers -predominantly Kaposi sarcoma and cervical cancer that were associated with late initiation of ART—WHO stage III and IV [44]. In this study, a low prevalence of cancer (0.7%) was found with cervical cancer as the predominant cancer type. The low prevalence of cancer could have been attributed to early initiation and test and treat ART policies. In this study only 13.2% of KPs had a CD4 of $<200$ cells/mm$^3$ at initiation of ART further explaining the low prevalence of AIDS-defining malignancies including Kaposi sarcoma. Additionally, introduction of cervical cancer screening towards the end of the study period in early 2015 could serve to explain the low number of cancer cases. Studies among HIV-infected FSWs in similar settings have found human papilloma virus (HPV) 51 and 52 showing independent associations with abnormal cervical cytology among FSWs [45]. Studies among MSM elsewhere found high rates of HPV type 16 infection that was associated with anal intraepithelial neoplasia (AIN) and anal cancer [46]. Records of cases in this study lacked both staging details and any associations with HPV serotypes thus limiting ability to further characterize the cancer burden.

This study represents some of the earliest attempts at quantifying NCD burden among KPs living with HIV in the SSA setting. However, this study was not without limitations. The cross-sectional nature of this study design limited inferences that could be adduced. While several studies, particularly from general population PLHIV demonstrate increased incident NCD cases [23, 30], this study's design limited the description of incidence, and outcomes following ART treatment.

Although this paper highlights the prevalence of key NCDs in a group of FSWs and MSM enrolled in a funded HIV prevention and treatment program in Nairobi Kenya, majority of those with conditions of interests were women. Similar to many other countries in SSA, sex work and same sex relationships in Kenya are not only generally criminalized but also highly stigmatized [20]. Females who sell sex however seem to be more tolerated than MSM engaged in the trade. The latter suffer double stigma and their position is further worsened by being HIV infected. Therefore, females were over represented in the study sample as fewer men out of the many closeted MSM have taken that leap of faith to enroll in ongoing HIV prevention and treatment programs. The over representation of women in this study's sample limits generalizability of study results to key populations experiences in Kenya or the region. That notwithstanding, this study provides important insights into NCD burden in this marginalized population that has not been reported elsewhere.

Several studies indicate high levels of discrimination and uneven access for KP to HIV/ NCD and SRH services [47–49]. However, being in an urban set-up and operating KP only services, with linkages to legal support systems, KPs in this study were considered emancipated with improved access to HIV care. The absence of routine glucose monitoring at SWOP clinics could have contributed to the absence of any reported cases of diabetes. Similarly, detection of hypertension based on updated guidelines that require a 24-hour mean blood pressure reading was limited owing to operational challenges at SWOP clinics [50].

## Conclusion

This study found a high prevalence of NCDs among KPs living with HIV on ART at a large prevention and treatment program in Nairobi Kenya. This study's results call for an urgent shift in refocusing HIV and NCD prevention in key populations targeted by ongoing programs in the face of a changing HIV epidemic. With integrated HIV/NCD care models being

considered to address the growing syndemic for general population PLHIV, KPs will require similar strategies. Efforts to operationalize HIV/NCD integration through strengthening workforce strategies and revision and simplification of HIV tools to include NCD screening are an urgent priority. Differentiated approaches to delivering KP services and overcoming of regulatory barriers to legitimize lay and peer approaches as part of healthcare system warrant consideration. Strengthening data collection and surveillance of NCDs among both general population and KP PLHIV are necessary to inform effective HIV/NCD integration prevention and treatment models and policies.

## Acknowledgments

We acknowledge all clients at all seven SWOP clinics, whose data was used in the preparation of this manuscript. Further, we acknowledge the University of Manitoba Research Group study team, who were instrumental in data collection activities. A sincere appreciation for all your support.

## Author Contributions

**Conceptualization:** Dunstan Achwoka, Joshua Kimani.

**Formal analysis:** Dunstan Achwoka, Patrick Munywoki, Thomas Achia.

**Investigation:** Dunstan Achwoka, Maureen Akolo, Joshua Kimani.

**Methodology:** Dunstan Achwoka, Patrick Munywoki, Thomas Achia.

**Project administration:** Joshua Kimani.

**Supervision:** Julius O. Oyugi, Thomas Achia, Festus Muriuki, Joshua Kimani.

**Validation:** Regina Mutave.

**Writing – original draft:** Dunstan Achwoka.

**Writing – review & editing:** Dunstan Achwoka, Julius O. Oyugi, Regina Mutave, Patrick Munywoki, Thomas Achia, Maureen Akolo, Festus Muriuki, Mercy Muthui, Joshua Kimani.

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
