## [Decision Letter · Decision Letter 0]

6 Apr 2020

PONE-D-19-32681

Noncommunicable disease burden among Key Population on Care and Treatment: a retrospective cross-sectional analysis of HIV-care outcomes from the Sex Workers Outreach Program in Kenya, 2012-2015

PLOS ONE

Dear Dr. Achwoka,

Thank you for submitting your manuscript to PLOS ONE. After careful consideration, we feel that it has merit but does not fully meet PLOS ONE’s publication criteria as it currently stands. Therefore, we invite you to submit a revised version of the manuscript that addresses the points raised during the review process.

We would appreciate receiving your revised manuscript by May 21 2020 11:59PM. To enhance the reproducibility of your results, we recommend that if applicable you deposit your laboratory protocols in protocols.io, where a protocol can be assigned its own identifier (DOI) such that it can be cited independently in the future. For instructions see: http://journals.plos.org/plosone/s/submission-guidelines#loc-laboratory-protocols

We look forward to receiving your revised manuscript.

Kind regards,

Joel Msafiri Francis, MD, MS, PhD

Academic Editor

PLOS ONE

Journal Requirements:

2. Please address the following:

- Please refer to any post-hoc corrections to correct for multiple comparisons during your statistical analyses. If these were not performed please justify the reasons. Please refer to our statistical reporting guidelines for assistance (https://journals.plos.org/plosone/s/submission-guidelines.#loc-statistical-reporting).

- Please modify the title to ensure that it is meeting PLOS’ guidelines (https://journals.plos.org/plosone/s/submission-guidelines#loc-title). In particular, the title should be "specific, descriptive, concise, and comprehensible to readers outside the field".

Thank you for your attention to these queries.

3. In ethics statement in the manuscript and in the online submission form, please provide additional information about the patient records used in your retrospective study. Specifically, please ensure that you have discussed whether all data were fully anonymized before you accessed them and/or whether the IRB or ethics committee waived the requirement for informed consent. If patients provided informed written consent to have data from their medical records used in research, please include this information.

5. We note that Figure 1 in your submission contain map images which may be copyrighted. All PLOS content is published under the Creative Commons Attribution License (CC BY 4.0), which means that the manuscript, images, and Supporting Information files will be freely available online, and any third party is permitted to access, download, copy, distribute, and use these materials in any way, even commercially, with proper attribution. For these reasons, we cannot publish previously copyrighted maps or satellite images created using proprietary data, such as Google software (Google Maps, Street View, and Earth). For more information, see our copyright guidelines: http://journals.plos.org/plosone/s/licenses-and-copyright.

Reviewers' comments:

Reviewer's Responses to Questions

**Comments to the Author**

1. Is the manuscript technically sound, and do the data support the conclusions?

Reviewer #1: Partly

Reviewer #2: Yes

Reviewer #3: Partly

Reviewer #4: Partly

2. Has the statistical analysis been performed appropriately and rigorously? 

Reviewer #1: Yes

Reviewer #2: Yes

Reviewer #3: Yes

Reviewer #4: Yes

3. Have the authors made all data underlying the findings in their manuscript fully available?

Reviewer #1: No

Reviewer #2: Yes

Reviewer #3: Yes

Reviewer #4: No

4. Is the manuscript presented in an intelligible fashion and written in standard English?

Reviewer #1: Yes

Reviewer #2: Yes

Reviewer #3: Yes

Reviewer #4: Yes

5. Review Comments to the Author

Reviewer #1: This paper show an interesting topic and information on NCD among key population living with HIV in Kenya. It shows a relatively high prevalence of NCDs in this population. The data is valuable but some aspects of the analysis and presentation of results are very unclear to me at present and require substantial improvement. I am also not convinced yet why the data cannot be share for publication with the reason explained. However, I found it is required a major revision with some initial comments and would like to see the revise version for additional comment.

Overall comments:

1- You may read the submission guideline of Plos One and adjust your manuscript organization as suggested such as page and line number, abbreviation used in abstract, and then resubmit it

2- I feel hard to give the specific comment because the manuscript miss the line number in page 1 to page 13

Specific comments:

1- I think the title of the paper is interesting. However, you may consider making it simpler and easier to catch up.

2- In your abstract you may consider to used less abbreviation as suggested by the submission guideline of Plos One

3- You used the team "HIV-infected Key Populations" in some lines it make reader confused. You may use the other term such as key population living with HIV or HIV positive key populations.

Reviewer #2: The research provides some interesting infos about NCDs burden in HIV+ KP.

Some minor revision to be made:

1. the abstract could be divided in sections to make it easier to read

2. among the limitations in the discussion section, maybe you could add something about the fact that it was not possible to diagnose hypertension according to international recommendations (3 measurements at the same time, etc...). It is clear that it was not possibile, but it would

3. I would suggest to add some other interesting papers to the references list, as:

a. Ibrahim MM, Damasceno A. Hypertension in developing countries. The Lancet. 2012;380(9841):611-9.

b. Mbanya JC, Squire S, Cazap E, Puska P. Mobilising the world for chronic NCDs. The Lancet. 2011;377(9765):536-7.

c. Bloomfield GS, Hogan JW, Keter A, Sang E, Carter EJ, Velazquez EJ, et al. Hypertension and Obesity as Cardiovascular Risk Factors among HIV Seropositive Patients in Western Kenya. PLOS ONE. 2011;6(7):e22288.

d. Ciccacci F, Tolno VT, Doro Altan A, Liotta G, Orlando S, Mancinelli S, et al. Non communicable diseases burden and risk factors in a cohort of HIV+ elderly patients in Malawi. AIDS Res Hum Retroviruses. 2019.

With this minor revision, I think the paper is valuable to be published, as provides new results for a particular group of patients that should receive more attention by the public health programs, also from and NCDs point of view.

Reviewer #3: This is great piece of work given the attention that HIV/NCD comorbidity is receiving currently especially in countries with high HIV burden undergoing rapid epidemiological transition. The manuscript presents a clear and transparent research process with results emanating from appropriate analyses.However, the author is advised to consider making the following changes to improve manuscript readability and technical soundness.

Comment 1: Table headings ought to be in uniform format. Table 1 and Table 2 headings appear to have inconsistent formats.

comment 2: Table 2 column N should be described either by a footnote or by column heading to avoid confusing the reader with another N=271. Table 3 footnotes should be numbered consecutively:1,2,3 and not 2,2,3.

Comment 3:Could you please make the last columns for Table 2 wider in order to cover confidence intervals in one line other than two lines as is the case now?This would help your Table 2 look tidier.

Comment 4:The sentence in line 53 could read better if you removed "albeit" and replaced "comparable" with "comparably".

Comment 5:Please interpret statistically significant odds ratios in univariate analyses.Also in line number 35,"unadjusted model" should be changed to unadjusted analyses or univariate analyses since this is not one model perse, all variables assessed in univariate analyses represent individual univariate logistic models.

Comment 6: A sentence in line 37 reads "When the model was adjusted, all prior significant associations between NCD diagnosis and increased age, unemployment status, BMI and CD4 ceased"

Consider changing this sentence to "even though increased age, unemployment status, BMI and CD4 were associated with NCD diagnosis in unadjusted analyses, they were not significantly associated with NCD diagnosis in the multivariate,adjusted analyses".The authors should also explain why variables with P> 0.2 such as sex, alcohol use and smoking were included in the multivariate model as this is not consistent with their analysis plan in which an automated stepwise backward logistic regression approach was selected to build a multivariate model to determine predictors for NCD prevalence from

independent predictors of NCDs with a p-value of 0.2 or less in univariate

analyses.

Comment 7: The author should consider adding references to his claim in line number 89 "Studies have

90 documented harmful consumption of alcohol, smoking tobacco, illicit drug use, and risky sexual behaviors

91 among KPs as factors that increase their risk for NCDs."

Comment 8: The author tries to contradict ealier research findings that they did not find an association between ART use and NCD prevalence in line number 66. This should be avoided as the study was not designed to show association between ART use and NCD prevalence since the study enrolled no participants without exposure to ART .On the same note,the author also reports lack of association between NCD and detected viral load.In this study, at no point were viral load measurements reported. Therefore such claims are not supported and should be removed from the manuscript.

Comment 8: The author should remove any references in the conclusion section of the manuscript.

Reviewer #4: The empirical analysis is competent, and the authors reference much of the relevant literature. A review on NCDs among key populations (KPs) is generally valuable, in light of concerns on the burden of NCDs among PLWH in general and specifically of the role and specific needs of key populations.

My reservations on publication of the paper in its present form primarily regard two aspects:

First, does the paper provide an analysis on NCDs among KPS? Not really. 94 percent of the sample are FSWs, and only 6 percent (n=86) MSMs. 86 percent of the cases of NCDs (total NCDs: n=271) are instances of high blood pressure (n=233), and the number instances of chronic respiratory disease (n=16) or cancers (n=10) do not allow a substantial empirical analysis. Against these numbers, is puzzling why much of the paper is cast in terms of "NCDs" among KPs – it really is about high blood pressure among FSWs, and – because implications and determinants of NCDs arguably differ – focusing the empirical analysis on instances of any NCD blurs the lessons which could be learned.

Second, are there any useful findings? Note sure – the results broadly mirror the empirical evidence on risk factors for hypertension – prevalence is increasing with age and with BMI. Other factors appear irrelevant (this could be tested more explicitly – do factors pervasive with respect to key populations play any role?). However, we would also want to understand whether prevalence of NCDs differs from prevalence in the general population. Doing such a comparison explicitly is beyond the scope of this paper (sample on KPs only), but the authors do not exhaust possibilities on comparing their findings with data on the general population (e.g., from DHS and related data). Relatedly, what is the relevance of the findings with regard to the management of HIV or NCDs among key populations?

Minor points:

It is not clear on what basis variables have been excluded in the multivariate analysis (Table 4). Excluding ART regimen and prior TB history appears sensible (p-value>0.9 in univariate regression), but there are numerous other variables with p-values in the vicinity or 0.7 or 0.8 in the multivariate analysis which are included in the regression.

Table 4: Review p-value of 1.11, BMI 30+ adjusted odds ratio.

In a couple of places, I felt that the paper would benefit from a round of copy-editing to improve precision.

6. PLOS authors have the option to publish the peer review history of their article (what does this mean?). If published, this will include your full peer review and any attached files.

Reviewer #1: Yes: Pheak CHHOUN

Reviewer #2: No

Reviewer #3: Yes: Blessings Gausi, MD MPH.

Reviewer #4: No

---

## [Author Response · Author response to Decision Letter 0]

24 Apr 2020

Reviewer #1: 

This paper shows an interesting topic and information on NCD among key population living with HIV in Kenya. It shows a relatively high prevalence of NCDs in this population. The data is valuable but some aspects of the analysis and presentation of results are very unclear to me at present and require substantial improvement. I am also not convinced yet why the data cannot be share for publication with the reason explained.

Response: We wish to thank the reviewer for this comment. We wish to confirm to the reviewer that we have made every effort to make our analysis and presentation of the data as transparent as possible. Following PLOS’s data privacy policy and declaration of Helsinki on protecting vulnerable populations, we are under ethical obligation to safeguard the identity of our study participants. Further, in Kenya, key populations are in conflict with the law, and efforts to recognize them have been vehemently thwarted. Under protocol P258/09/2008, the Kenyatta National Hospital – University of Nairobi (KNH-UON) Ethics Research Committee has imposed restriction on the access of this data citing that sharing would be deemed to increase the risks or affect the safety or welfare of study participants. However, we wish to confirm that data access can be granted upon request to the Secretary at KNH-UoN Ethics and Research Committee (uonknh_erc@uonbi.ac.ke) for researchers who meet the criteria for access to confidential data.

Overall comments:

Reviewer Comment 1- You may read the submission guideline of Plos One and adjust your manuscript organization as suggested such as page and line number, abbreviation used in abstract, and then resubmit it

Response: We thank the reviewer for making this observation. In our resubmission, we have carefully followed PLOS One’s submission guidelines to reformat the manuscript, addressing its organization, page and line numbers. We have also spelled out abbreviation used in the abstract at the first instance that they are used.

Reviewer Comment 2- I feel hard to give the specific comment because the manuscript misses the line number in page 1 to page 13

Response: We have since reformatted the resubmitted manuscript and now have line numbers for pages 1 to 13.

Specific comments:

Reviewer Comment 1- I think the title of the paper is interesting. However, you may consider making it simpler and easier to catch up.

Response: We take note of the reviewer’s suggestion. Our updated title now reads “High prevalence of noncommunicable diseases among Key populations enrolled at a large HIV prevention and treatment program in Kenya”.

Reviewer Comment 2- In your abstract you may consider to used less abbreviation as suggested by the submission guideline of Plos One

Response: As per the reviewer’s comment and Plos One submission guideline, we have reduced the number of abbreviations in the abstract. Additionally, we have spelled out each abbreviation at the first instance it is used.

Reviewer Comment 3- You used the team "HIV-infected Key Populations" in some lines it make reader confused. You may use the other term such as key population living with HIV or HIV positive key populations.

Response: We apologize for any confusion that may have arisen as a result of using the term “HIV-infected Key populations”. We have since updated the manuscript using the term “key populations living with HIV”.

Reviewer #2:

Minor revisions to be made:

Reviewer Comment 1. The abstract could be divided in sections to make it easier to read

Response: We have since divided the abstract into sections improving its readability.

Reviewer Comment 2. Among the limitations in the discussion section, maybe you could add something about the fact that it was not possible to diagnose hypertension according to international recommendations (3 measurements at the same time, etc...).

Response: We thank the reviewer for this suggestion. We have included a sentence and a reference to reflect this limitation. The sentence (lines 358-359) reads “Similarly, our detection of hypertension based on updated guidelines that require a 24-hour mean blood pressure reading was limited owing to operational challenges at SWOP clinics .”

3. I would suggest to add some other interesting papers to the references list, as:

a. Ibrahim MM, Damasceno A. Hypertension in developing countries. The Lancet. 2012;380(9841):611-9.

b. Mbanya JC, Squire S, Cazap E, Puska P. Mobilising the world for chronic NCDs. The Lancet. 2011;377(9765):536-7.

c. Bloomfield GS, Hogan JW, Keter A, Sang E, Carter EJ, Velazquez EJ, et al. Hypertension and Obesity as Cardiovascular Risk Factors among HIV Seropositive Patients in Western Kenya. PLOS ONE. 2011;6(7):e22288.

d. Ciccacci F, Tolno VT, Doro Altan A, Liotta G, Orlando S, Mancinelli S, et al. Non communicable diseases burden and risk factors in a cohort of HIV+ elderly patients in Malawi. AIDS Res Hum Retroviruses. 2019.

Response: We thank the reviewer for the suggested references. We have since updated our references with three of the four journal articles.

 

Reviewer #3:

This is great piece of work given the attention that HIV/NCD comorbidity is receiving currently especially in countries with high HIV burden undergoing rapid epidemiological transition. The manuscript presents a clear and transparent research process with results emanating from appropriate analyses. However, the author is advised to consider making the following changes to improve manuscript readability and technical soundness.

Reviewer Comment 1: Table headings ought to be in uniform format. Table 1 and Table 2 headings appear to have inconsistent formats.

Response: We thank the reviewer for their comments. We have formatted the table headings and now have a consistent format from Table 1 to 4.

Reviewer comment 2: Table 2 column N should be described either by a footnote or by column heading to avoid confusing the reader with another N=271. Table 3 footnotes should be numbered consecutively:1,2,3 and not 2,2,3.

Response: We have updated the column labelling to consistently reflect a common “N” that is unambiguous. We have also updated the consecutive numbering of footnotes on Table 3. 

Reviewer Comment 3: Could you please make the last columns for Table 2 wider in order to cover confidence intervals in one line other than two lines as is the case now? This would help your Table 2 look tidier.

Response: We oblige and have reformatted Table 2 to cover confidence intervals in one line.

Reviewer Comment 4: The sentence in line 53 could read better if you removed "albeit" and replaced "comparable" with "comparably".

Response: As per the reviewer’s suggestion, we have removed the word “albeit” on line 53 and replaced “comparable” with “comparably” to improve its readability.’

Reviewer Comment 5: Please interpret statistically significant odds ratios in univariate analyses. Also in line number 35,"unadjusted model" should be changed to unadjusted analyses or univariate analyses since this is not one model perse, all variables assessed in univariate analyses represent individual univariate logistic models.

Response: Thank you for pointing this out. We have provided an interpretation of the significant odds ratios from univariate analysis and now refer to univariate analyses in line 256 - 257 as opposed to “unadjusted model”.

Reviewer Comment 6: A sentence in line 37 reads "When the model was adjusted, all prior significant associations between NCD diagnosis and increased age, unemployment status, BMI and CD4 ceased" Consider changing this sentence to "even though increased age, unemployment status, BMI and CD4 were associated with NCD diagnosis in unadjusted analyses, they were not significantly associated with NCD diagnosis in the multivariate, adjusted analyses".

Response: As per the reviewer’s suggestion, we have since updated the sentence in line 258 to read as follows " Even though increased age, BMI and CD4 were associated with NCD diagnosis in unadjusted analyses, significant association with NCD diagnosis in the adjusted analyses remained only for categories of BMI 30 kg/m2 and above, and ages 45 years and above (borderline statistically significant)".

Reviewer Comment 6 part B: The authors should also explain why variables with P> 0.2 such as sex, alcohol use and smoking were included in the multivariate model as this is not consistent with their analysis plan in which an automated stepwise backward logistic regression approach was selected to build a multivariate model to determine predictors for NCD prevalence from independent predictors of NCDs with a p-value of 0.2 or less in univariate analyses.

Response: Apologies for the deficiency in description of our methods. Age, sex, alcohol use and smoking were considered apriori as potential confounders of the association of studied risk factors with NCD and included in the final multivariate logistic regression model. We have updated the statistical analysis section accordingly, line 118-119. 

Reviewer Comment 7: The author should consider adding references to his claim in line number 89 "Studies have [90] documented harmful consumption of alcohol, smoking tobacco, illicit drug use, and risky sexual behaviors [91] among KPs as factors that increase their risk for NCDs."

Response: As per the reviewer’s comment, we have included three references that support the claim on documented harmful consumption of alcohol, smoking tobacco, illicit drug use, and risky sexual behaviors [91] among KPs as factors that increase their risk for NCDs. 

Reviewer Comment 8: The author tries to contradict earlier research findings that they did not find an association between ART use and NCD prevalence in line number [66]. This should be avoided as the study was not designed to show association between ART use and NCD prevalence since the study enrolled no participants without exposure to ART. On the same note, the author also reports lack of association between NCD and detected viral load. In this study, at no point were viral load measurements reported. Therefore, such claims are not supported and should be removed from the manuscript.

Response: We thank the reviewer for this suggestion and have expunged the line 66 at the end of paragraph two in the discussion section. Indeed, our study did not enroll participants with no ART exposure therefore unable to make the claim of an association between ART use and NCD prevalence. We have also removed any mention of an association with viral load measurements in our study since these were not presented in this manuscript.

Reviewer Comment 9: The author should remove any references in the conclusion section of the manuscript.

Response: We have removed all references in the conclusion section.

Reviewer #4:

The empirical analysis is competent, and the authors reference much of the relevant literature. A review on NCDs among key populations (KPs) is generally valuable, in light of concerns on the burden of NCDs among PLWH in general and specifically of the role and specific needs of key populations.

My reservations on publication of the paper in its present form primarily regard two aspects:

First, does the paper provide an analysis on NCDs among KPS? Not really. 94 percent of the sample are FSWs, and only 6 percent (n=86) MSMs. 86 percent of the cases of NCDs (total NCDs: n=271) are instances of high blood pressure (n=233), and the number instances of chronic respiratory disease (n=16) or cancers (n=10) do not allow a substantial empirical analysis. Against these numbers, is puzzling why much of the paper is cast in terms of "NCDs" among KPs – it really is about high blood pressure among FSWs, and – because implications and determinants of NCDs arguably differ – focusing the empirical analysis on instances of any NCD blurs the lessons which could be learned.

Response: Thank you for the thoughtful comment. We utilized data from a routine HIV prevention and treatment program serving the two key populations. As such, the distribution of attended population and outcomes of interest was analyzed as-is. The reviewers rightly points out this and as part of the paper we now acknowledge this limitation in the discussion and have reframed our interpretation in this light. Although the paper highlights the prevalence of key NCDs in group of female and male sex workers enrolled in a funded HIV prevention and treatment program in Nairobi Kenya, majority of those with conditions of interests were women. Sex work and same sex relationships are still illegal and stigmatized in Kenya. However, females who sell sex seem to be more tolerated than MSM engaged in the trade. The latter suffer double stigma and their position is even worsened by being HIV infected. Therefore, females were over represented in the study sample as fewer men out of the many closeted MSM have taken that leap of faith to enroll in the ongoing HIV prevention and treatment programs. We acknowledge that the over representation of women in the sample limits the generalizability of results to key populations experiences in Kenya or the region. However, the data provides some insights into NCD burden in this marginalized population that has not been reported elsewhere. 

Second, are there any useful findings? Note sure – the results broadly mirror the empirical evidence on risk factors for hypertension – prevalence is increasing with age and with BMI. Other factors appear irrelevant (this could be tested more explicitly – do factors pervasive with respect to key populations play any role?). However, we would also want to understand whether prevalence of NCDs differs from prevalence in the general population. Doing such a comparison explicitly is beyond the scope of this paper (sample on KPs only), but the authors do not exhaust possibilities on comparing their findings with data on the general population (e.g., from DHS and related data). Relatedly, what is the relevance of the findings with regard to the management of HIV or NCDs among key populations?

Response: Again, we greatly appreciate this insightful comment. While we do acknowledge the limitations of our study in this regard, we note that our recommendation to funders and policy makers to promote integration of NCD- HIV programming in the future is evidence based too. We have previously published NCD prevalence’s from the general population using similar routine HIV prevention and treatment data (ref Achwoka D, Waruru A, Chen T-H, Masamaro K, Ngugi E, Kimani M, et al. Noncommunicable disease burden among HIV patients in care: a national retrospective longitudinal analysis of HIV-treatment outcomes in Kenya, 2003-2013. 2019;19(1):372. doi: 10.1186/s12889-019-6716-2.). PLHIV in Kenya were found to have a high prevalence of NCD diagnoses with proportion of any documented NCD among PLHIV being 11.5% (95% confidence interval [CI] 9.3, 14.1). 

Minor points:

It is not clear on what basis variables have been excluded in the multivariate analysis (Table 4). Excluding ART regimen and prior TB history appears sensible (p-value>0.9 in univariate regression), but there are numerous other variables with p-values in the vicinity or 0.7 or 0.8 in the multivariate analysis which are included in the regression.

Response: Apologies for the deficiency in description of our methods. Age, sex, alcohol use and smoking were considered apriori as potential confounders of the association of studied risk factors with NCD and included in the final multivariate logistic regression model. We have updated the statistical analysis section accordingly.

Table 4: Review p-value of 1.11, BMI 30+ adjusted odds ratio.

Response: This was an error. The correct p-value is 0.029. 

In a couple of places, I felt that the paper would benefit from a round of copy-editing to improve precision.

Response: We have revised the whole manuscript following PLOS One guideline and making the presentation as succinct as possible. The revised manuscript has a couple edits to that effect.

---

## [Decision Letter · Decision Letter 1]

27 May 2020

PONE-D-19-32681R1

High prevalence of noncommunicable diseases among Key populations enrolled at a large HIV prevention and treatment program in Kenya

PLOS ONE

Dear Dr. Achwoka,

Thank you for submitting your manuscript to PLOS ONE. After careful consideration, we feel that it has merit but does not fully meet PLOS ONE’s publication criteria as it currently stands. Therefore, we invite you to submit a revised version of the manuscript that addresses the points raised during the review process.

We look forward to receiving your revised manuscript.

Kind regards,

Joel Msafiri Francis, MD, MS, PhD

Academic Editor

PLOS ONE

Reviewers' comments:

Reviewer's Responses to Questions

**Comments to the Author**

1. If the authors have adequately addressed your comments raised in a previous round of review and you feel that this manuscript is now acceptable for publication, you may indicate that here to bypass the “Comments to the Author” section, enter your conflict of interest statement in the “Confidential to Editor” section, and submit your "Accept" recommendation.

Reviewer #1: (No Response)

Reviewer #2: All comments have been addressed

Reviewer #4: (No Response)

2. Is the manuscript technically sound, and do the data support the conclusions?

Reviewer #1: Yes

Reviewer #2: Yes

Reviewer #4: Yes

3. Has the statistical analysis been performed appropriately and rigorously? 

Reviewer #1: Yes

Reviewer #2: Yes

Reviewer #4: Yes

4. Have the authors made all data underlying the findings in their manuscript fully available?

Reviewer #1: No

Reviewer #2: No

Reviewer #4: No

5. Is the manuscript presented in an intelligible fashion and written in standard English?

Reviewer #1: No

Reviewer #2: Yes

Reviewer #4: Yes

6. Review Comments to the Author

Reviewer #1: Many thanks for your revision. I found a great improvement with the revised version. To move on further process, I am suggesting for a few more minor revisions as described below;

1- I think you should remove capital letter of “Key populations” in the title and a few other places in the manuscript

2- I think you should add in text citation and reference for first sentence in the second paragraph (line 51-53).

3- I think you should use less human possessive term in the manuscript. You should make the language use to more academic. You should consider removing those words such as in line 72 “we only…”, line 107 “Our analyses”, line 136 “We analyzed”, line 216 “Our proxy”, line 270 “our study”, line 271 “In our study”, line 274 “ours among”, line 286, line 289, “in our study”, line 295 “we…” … “our”… please check for the rest and you may use term “this study” to replace “our study”…

4- Do you have any rational why age, gender, alcohol use and smoking were considered as priori as potential confounder? Any learning from other literature?

5- In line 274, you mention “studies similar to ours among…” it seems does not fully completed yet.

6- At the end of the data collection or the starting of data analysis, you may add another sentence indicating that data is cleaned and imported into Stata for data analysis.

7- In the data analysis, you may need to also describe how will you report the result of of the analysis such mean, median, SD, 95% CI… and abbreviate any possible term here than you don’t have to write full word in the result (eg. in line 160, 95% confident interval (CI)).

8- In line 136-138, I think this seem reported the data collection and analysis section. You may start directly reporting the result of analysis to be concise.

9- In line 139, I think you can use abbreviation of FSWs and MSM as they been abbreviated already in the introduction. You may also need to check other line to make it consistent.

10- Could you please add concrete list of inclusion / exclusion criteria for administer abstracted data? I found pieces of information from line 85 to 90 but not so convinced yet.

11- I still would like to suggest, the author consider to discuss their finding with the other studies such as “High prevalence of non-communicable diseases and associated risk factors amongst adults living with HIV in Cambodia” and/or “Non-communicable diseases and related risk behaviors among men and women living with HIV in Cambodia: Findings from a cross-sectional study” because I found this study is quite similar in some setting this study participant are people living with HIV even it focus to key population.

Reviewer #2: The authors considered and addressed all the comments. The concerns related to the availability of data have been clarified.

Reviewer #4: The authors provide competent and diligent responses to reviewers' comments, specifically those by myself but also (according to a cursory overview) those from other reviewers.

One - I believe - important shortcoming remains. The authors do not provide a substantial discussion comparing NCDs among key populations (KPs) with the prevalence of NCDs in the general population or among PLWH overall. Understanding these differences, though, would be important for interpreting the findings and drawing policy-relevant conclusions. Are there factors specific to KPs driving high prevalence of NCDs? The findings suggest that this may not be the case, as pointers for risk behaviour come out largely insignificant (the statistically significant variables are age and BMI 30+). A pointer to the findings of the paper referred to in the response to comments (Noncommunicable disease burden among HIV patients in care: a national retrospective longitudinal analysis...), with some author overlap with the present paper, would also contribute to placing the findings of the present paper in context, and I find it puzzling to see that the authors do not make such connections.

This shortcoming may not preclude publication, as the analysis per se is competent, but the value of the paper is clearly diminished by the authors' reluctance to place their findings in this wider population or PLWH context.

On the editorial side, I sense that while the language is clear throughout, the paper would benefit from one round of professional copy-editing.

7. PLOS authors have the option to publish the peer review history of their article (what does this mean?). If published, this will include your full peer review and any attached files.

Reviewer #1: Yes: Pheak Chhoun

Reviewer #2: No

Reviewer #4: No

---

## [Author Response · Author response to Decision Letter 1]

30 May 2020

Response to Reviewers' Comments:

Reviewer #1: 

Many thanks for your revision. I found a great improvement with the revised version. To move on further process, I am suggesting for a few more minor revisions as described below;

1- I think you should remove capital letter of “Key populations” in the title and a few other places in the manuscript

Response: We thank the reviewer for this suggestion. We have since removed the capital letter ‘K’ in key populations both in the title and in the manuscript text.

2- I think you should add in text citation and reference for first sentence in the second paragraph (line 51-53).

Response: As suggested by the reviewer, we have included an in-text citation and reference for the first sentence in the second paragraph.

3- I think you should use less human possessive term in the manuscript. You should make the language use to more academic. You should consider removing those words such as in line 72 “we only…”, line 107 “Our analyses”, line 136 “We analyzed”, line 216 “Our proxy”, line 270 “our study”, line 271 “In our study”, line 274 “ours among”, line 286, line 289, “in our study”, line 295 “we…” … “our”… please check for the rest and you may use term “this study” to replace “our study”…

Response: We continue to thank the reviewer for this important comment. The entire manuscript has been updated accordingly to rid it of human possessive terms and is now entirely in academic language. 

4- Do you have any rational why age, gender, alcohol use and smoking were considered as priori as potential confounder? Any learning from other literature?

Response: We thank the reviewer for this query. The aforementioned factors met the criteria for confounding (being associated with both the risk factor of interest and the outcome, unequal distribution among comparison groups, and not being an intermediary step in causal pathway). We found these factors in our literature review and point the reviewer to the Kenya STEPwise survey for non-communicable risk factors 2015 Report. 

5- In line 274, you mention “studies similar to ours among…” it seems does not fully completed yet.

Response: We realize that line 274 was unclear and have revised line 273-275 to improve clarity. It now reads “ Despite a heightened impetus to refocus on populations at increased risk for both NCDs and HIV infection , studies among key populations living with HIV remain rare [22]. That notwithstanding, systematic reviews outside SSA suggest that sexual minorities exhibit higher rates of NCDs [11]”.

6- At the end of the data collection or the starting of data analysis, you may add another sentence indicating that data is cleaned and imported into Stata for data analysis.

Response: We have incorporated this important suggestion in the manuscript under the sub-title ‘Study procedures and data collection’. It now reads “Data were cleaned and subsequently imported to STATA 15 (STATA Corporation, Texas USA) for data analysis”.

7- In the data analysis, you may need to also describe how will you report the result of of the analysis such mean, median, SD, 95% CI… and abbreviate any possible term here than you don’t have to write full word in the result (eg. in line 160, 95% confident interval (CI)).

Response: We thank the reviewer for this comment. We have now included a sentence within the statistical analyses to describe reporting of analyses such as mean, SD and 95% confidence intervals.

8- In line 136-138, I think this seem reported the data collection and analysis section. You may start directly reporting the result of analysis to be concise.

Response: We agree with the reviewer’s suggestion. We have since revised the lines 136-138 to avoid repetition with the data collection and analysis section. It is now concise.

9- In line 139, I think you can use abbreviation of FSWs and MSM as they been abbreviated already in the introduction. You may also need to check other line to make it consistent.

Response: We have updated the entire manuscript and checked for consistency as per the reviewer’s comment. Now the abbreviations of FSWs and MSM appear consistently.

10- Could you please add concrete list of inclusion / exclusion criteria for administer abstracted data? I found pieces of information from line 85 to 90 but not so convinced yet.

Response: We have bolstered this section as suggested by the reviewer and included both inclusion and exclusion criteria for the data abstraction. As part of inclusion we considered: a) age 15 and above; b) Enrollment into the SWOP clinics between periods October 2012 and September 2015; c) HIV positive at enrollment or seroconverted during the period of study and d) Identified as MSM or FSW as typology. For exclusion we considered: a) HIV negative key population; b) other key population typology – including PWIDs and transgender; c) key population enrolled outside the study period and d) key population under the age of 15 or missing information on age.

11- I still would like to suggest, the author consider to discuss their finding with the other studies such as “High prevalence of non-communicable diseases and associated risk factors amongst adults living with HIV in Cambodia” and/or “Non-communicable diseases and related risk behaviors among men and women living with HIV in Cambodia: Findings from a cross-sectional study” because I found this study is quite similar in some setting this study participant are people living with HIV even it focus to key population.

Response: We thank the reviewer for this insightful suggestion. We note that we have taken due diligence and studied the two papers. We are happy to report that the findings from the two papers from Cambodia add value to our manuscript and have been included as part of our discussion. Specifically, they are referenced in the second paragraph in the discussion.

Reviewer #4:

Minor revisions to be made:

Reviewer Comment 1. The authors provide competent and diligent responses to reviewers' comments, specifically those by myself but also (according to a cursory overview) those from other reviewers.

One - I believe - important shortcoming remains. The authors do not provide a substantial discussion comparing NCDs among key populations (KPs) with the prevalence of NCDs in the general population or among PLWH overall. Understanding these differences, though, would be important for interpreting the findings and drawing policy-relevant conclusions. Are there factors specific to KPs driving high prevalence of NCDs? The findings suggest that this may not be the case, as pointers for risk behaviour come out largely insignificant (the statistically significant variables are age and BMI 30+). A pointer to the findings of the paper referred to in the response to comments (Noncommunicable disease burden among HIV patients in care: a national retrospective longitudinal analysis...), with some author overlap with the present paper, would also contribute to placing the findings of the present paper in context, and I find it puzzling to see that the authors do not make such connections.

This shortcoming may not preclude publication, as the analysis per se is competent, but the value of the paper is clearly diminished by the authors' reluctance to place their findings in this wider population or PLWH context.

Response: We are thankful to the reviewer for raising this important concern. We acknowledge the gravity of the reviewer’s sentiments and have bolstered our discussion substantially to include comparisons of NCDs between key populations and general population PLHIV. We have included a new paragraph in the discussion section (now appears as the second paragraph) and added three references. These references include studies from concentrated HIV epidemics in Cambodia, recent modeling of NCD burden among PLHIV and general population as well as our very own study among general population PLHIV. We believe this will offer justice and improve the relevance of our study for policy conclusions.

Reviewer Comment 2. On the editorial side, I sense that while the language is clear throughout, the paper would benefit from one round of professional copy-editing.

Response: We thank the reviewer for this suggestion. We have conducted additional copy editing and would like to assure both the reviewer and editor that we now have a high-quality product commensurate to PLOS one’s standards.

---

## [Decision Letter · Decision Letter 2]

19 Jun 2020

High prevalence of non-communicable diseases among key populations enrolled at a large HIV prevention and treatment program in Kenya

PONE-D-19-32681R2

Dear Dr. Achwoka,

We’re pleased to inform you that your manuscript has been judged scientifically suitable for publication and will be formally accepted for publication once it meets all outstanding technical requirements.

Kind regards,

Joel Msafiri Francis, MD, MS, PhD

Academic Editor

PLOS ONE

Additional Editor Comments (optional):

Reviewers' comments:

Reviewer's Responses to Questions

**Comments to the Author**

1. If the authors have adequately addressed your comments raised in a previous round of review and you feel that this manuscript is now acceptable for publication, you may indicate that here to bypass the “Comments to the Author” section, enter your conflict of interest statement in the “Confidential to Editor” section, and submit your "Accept" recommendation.

Reviewer #1: All comments have been addressed

Reviewer #2: All comments have been addressed

Reviewer #4: All comments have been addressed

2. Is the manuscript technically sound, and do the data support the conclusions?

Reviewer #1: Yes

Reviewer #2: (No Response)

Reviewer #4: Yes

3. Has the statistical analysis been performed appropriately and rigorously? 

Reviewer #1: Yes

Reviewer #2: (No Response)

Reviewer #4: Yes

4. Have the authors made all data underlying the findings in their manuscript fully available?

Reviewer #1: Yes

Reviewer #2: (No Response)

Reviewer #4: No

5. Is the manuscript presented in an intelligible fashion and written in standard English?

Reviewer #1: Yes

Reviewer #2: (No Response)

Reviewer #4: Yes

6. Review Comments to the Author

Reviewer #1: I would like to thank the authors for considering the comments and revised the entires manuscript according. I have any specific comment to the manuscript in this round. Well done and should be off for the hard work.

Reviewer #2: (No Response)

Reviewer #4: The draft now reads much better, thanks also to the very thorough and specific comments from reviewer #1, and is fit to see the light of day.

7. PLOS authors have the option to publish the peer review history of their article (what does this mean?). If published, this will include your full peer review and any attached files.

Reviewer #1: No

Reviewer #2: No

Reviewer #4: No

---

## [Editor Report · Acceptance letter]

23 Jun 2020

PONE-D-19-32681R2 

High prevalence of non-communicable diseases among key populations enrolled at a large HIV prevention & treatment program in Kenya 

Dear Dr. Achwoka:

I'm pleased to inform you that your manuscript has been deemed suitable for publication in PLOS ONE. Congratulations! Your manuscript is now with our production department. 

Kind regards, 

on behalf of

Dr. Joel Msafiri Francis 

Academic Editor

PLOS ONE